# Estimation of Greenhouse Lettuce Growth Indices Based on a Two-Stage CNN Using RGB-D Images

**DOI:** 10.3390/s22155499

**Published:** 2022-07-23

**Authors:** Min-Seok Gang, Hak-Jin Kim, Dong-Wook Kim

**Affiliations:** 1Department of Biosystems Engineering, College of Agriculture and Life Sciences, Seoul National University, Seoul 08826, Korea; msg1907@snu.ac.kr (M.-S.G.); dwk8033@snu.ac.kr (D.-W.K.); 2Integrated Major in Global Smart Farm, College of Agriculture and Life Sciences, Seoul National University, Seoul 08826, Korea; 3Research Institute of Agriculture and Life Sciences, Seoul National University, Seoul 08826, Korea

**Keywords:** convolutional neural network (CNN), growth estimation, growth index, growth monitoring, greenhouse, lettuce, red, green, blue, and depth (RGB-D), stereo camera

## Abstract

Growth indices can quantify crop productivity and establish optimal environmental, nutritional, and irrigation control strategies. A convolutional neural network (CNN)-based model is presented for estimating various growth indices (i.e., fresh weight, dry weight, height, leaf area, and diameter) of four varieties of greenhouse lettuce using red, green, blue, and depth (RGB-D) data obtained using a stereo camera. Data from an online autonomous greenhouse challenge (Wageningen University, June 2021) were employed in this study. The data were collected using an Intel RealSense D415 camera. The developed model has a two-stage CNN architecture based on ResNet50V2 layers. The developed model provided coefficients of determination from 0.88 to 0.95, with normalized root mean square errors of 6.09%, 6.30%, 7.65%, 7.92%, and 5.62% for fresh weight, dry weight, height, diameter, and leaf area, respectively, on unknown lettuce images. Using red, green, blue (RGB) and depth data employed in the CNN improved the determination accuracy for all five lettuce growth indices due to the ability of the stereo camera to extract height information on lettuce. The average time for processing each lettuce image using the developed CNN model run on a Jetson SUB mini-PC with a Jetson Xavier NX was 0.83 s, indicating the potential for the model in fast real-time sensing of lettuce growth indices.

## 1. Introduction

Fresh weight, dry weight, height, diameter, and leaf area are growth indicators of farmland and horticultural crops because they are closely associated with transpiration and photosynthesis [1,2,3]. The accurate measurement of these indices enables us to quantify and model crop growth for bioenvironmental factors. Climate, nutrient, and irrigation control strategies can be established based on well-developed crop models to obtain optimal productivity. Growth indices can also help farmers determine appropriate transplanting or harvesting times to increase their economic status. These applications of growth index measurements may be even more effective, especially in greenhouses, where strict environmental controls are possible.

Traditional methods for measuring growth indicators are costly because operators typically collect plant samples and use analytical instruments in a laboratory. In addition, the reliability of the observed values depends on the sampling methods and sample conditions [4]. Some indices can be calculated from physical traits, such as leaf length and width [5,6]. However, direct manual measurements are still required, and the nonlinear relationship between traits can negatively affect model accuracy. Therefore, methods that estimate growth indices using sensors and crop image data have been effective alternatives to traditional methods for many researchers. These methods allow for the efficient collection of various indices [7,8,9,10].

Recently, the focus has been on deep learning techniques to automatically extract and use key features from numerous images. Deep neural networks are advantageous for solving nonlinear problems and can learn a relational model based on input and output through multiple hidden layers. Deep learning techniques do not require handcrafted features selected by experts and provide a relatively fast processing speed in inference compared to traditional computer vision-based methods [11]. Research has been conducted on crop species classification [12,13,14], plant and fruit localization [14,15], and pest diagnosis [16] by applying deep learning techniques. Studies have also been conducted on estimating quantitative growth indicators [14,17,18,19,20,21,22] and classifying or detecting objects.

Several researchers have applied segmentation using deep learning techniques to specify crop areas from red, green, blue (RGB) and three-dimensional (3D) data to estimate crop growth indices [18,19,20,21]. However, only the linear regression, random forest, and simple artificial neural network (ANN) models were used to estimate growth indices using features extracted from the segmented area. There are potential limitations in expressing the relationship between growth indicators and various features, such as color and texture. In addition, the segmentation was computationally costly and could increase image processing time, making it difficult to apply to real-time agricultural applications.

In this regard, a regression using a convolutional neural network (CNN) can be an alternative. The CNN has advantages in image processing and can reduce the processing time by effectively learning image features through kernel structures in the form of a 2D array. Zhang et al. developed a CNN model using RGB images and estimated the leaf fresh weight, leaf dry weight, and leaf area of greenhouse lettuce. They demonstrated that the estimation of these indicators could be superior to the estimation in the support vector machine, random forest, and linear regression models [22].

In addition, 3D data, such as RGB-D images, may provide more information on vertical changes in leaves and stems [23,24,25,26]. Quan et al. developed a novel two-stream CNN model based on RGB-D images to estimate the fresh weight of aboveground weeds in a field on a high-end graphics processing unit (GPU) (2080Ti, NVIDIA, Santa Clara, CA, USA) [27]. The two-stream CNN model used a multi-input single-output (MISO) structure and dense network in the network blocks. Quan et al. revealed that using depth images and a CNN effectively estimated the vertical changes as weeds grow. However, the dataset consisted of weeds with small-scale fresh weights and relatively short growth periods.

Several researchers used the public RGB-D dataset of greenhouse lettuces [28,29] published by Hemming et al. [30], whose dataset was built for the online challenge session of the Third Autonomous Greenhouse Challenge. Raja et al. developed deep learning models, including multi-input multi-output (MIMO) and MISO models, using a CNN and a deformable CNN (DCNN) to estimate growth indices [28]. Raja et al. reported a normalized mean squared error (NMSE) of 0.068 for MIMO and 0.069 for MISO. Li and Wang generated point cloud data from the same RGB-D dataset [30] and estimated the height of lettuce through a CNN and recurrent neural network [29].

Nevertheless, developing a CNN model using 3D data to estimate growth indices more effectively was necessary due to relatively poor performance in predicting the biophysical properties of lettuces in the late growth stage. It is important to investigate model applicability over the entire growth period of harvestable crops with quantitative comparisons. In many crops, vertical growth increases toward the later growth stages [31,32,33]. Moreover, [31,32,33] found that the height of the leafy and fruit vegetables increases quadratically or exponentially in the late growth stages until saturation. The estimation accuracy for the late stage may affect the performance of deep learning models [34]. In addition, the attainment of real-time processing speed on the embedded board should also be considered for agricultural applications of the neural network model that can handle various growth indicators.

Studies using the CNN and RGB-D models have applied MISO and MIMO structures. A two-stage architecture connects the two individual models in series and may extend and improve these structures. Two-stage architecture might be effective for data with time-series characteristics [30,35,36,37,38].

In this study, a two-stage CNN model was presented using RGB-D data to estimate various growth indicators (i.e., fresh weight, dry weight, height, diameter, and leaf area) of greenhouse lettuces in real time. In particular, the model estimation performance for the entire growth period was quantitatively presented. The RGB-D images published by Heming et al. [30] for four different varieties of lettuces (i.e., Lugano, Salanova, Aphylion, and (Satine) were used. The coefficient of determination (R^2^), root mean square error (RMSE), and normalized RMSE (NRMSE) were determined to evaluate the performance of the developed model for each growth index. In addition, the processing speeds per image were analyzed to investigate the real-time performance of the developed model on an embedded board.

## 2. Materials and Methods

### 2.1. Dataset

The dataset in this study was originally produced by Hemming et al. [30] for the online challenge session of the Third Autonomous Greenhouse Challenge. The dataset is publicly available at the 4TU.ResearchData website and consists of 388 RGB images, depth images, and actual aboveground data corresponding to a pair of RGB images and depth images, respectively. The RGB images are 24-bit portable network graphic (PNG) images with three channels, and the depth images are 8-bit PNG images with a single channel. The resolution of all the images is 1080 × 1920 pixels (Figure 1).

An Intel camera (RealSense D415, Intel, Santa Clara, CA, USA) was used for image acquisition. The D415 is equipped with RGB and stereo cameras to extract depth information. In principle, the D415 calculates depth information using the difference in the distance between the corresponding points of the left and right camera images. The data were collected by fixing the camera 0.9 m from the top of the plant through a metal frame to photograph the plant from top to bottom, maintaining the same shooting height and resolution.

In the dataset, four varieties of lettuce, namely Lugano, Salanova, Aphylion, and Satine, were photographed. The number of image pairs for each variety was 96, 102, 92, and 98, respectively. The plants were hydroponically grown in an experimentally controlled greenhouse in Bleiswijk, Netherlands. Data were acquired weekly from approximately 70 sampled plants at each growth stage for seven weeks from transplanting the seedlings until the crops reached a harvestable size.

Shoot fresh weight, shoot dry weight, height, diameter, and leaf area were obtained using a destructive measurement method (Table 1). Fresh weight was obtained by measuring the weight of lettuce harvested from the point where the first leaf was attached, and dry weight was measured after 3 days of drying in an oven after obtaining the fresh weight. Leaf area was calculated through the surface area projected onto a plane after separating the leaves from the stem, and the increase in leaf area due to leaf curvature was not considered. The diameter of the lettuce projected onto a plane was measured, and the height was measured from where the first leaf was attached to the highest point of the plant. Table 2 lists the range of each data measurement for the lettuce varieties.

Figure 2 illustrates an example of the lettuce depth image data. The depth images were aligned to have the same pixel area as the RGB images. As the value of each pixel in the depth image data was between 681 and 1306, it was difficult to visualize the depth data directly. Therefore, the pixel values were normalized using the maximum and minimum values of Equation (1) and were multiplied by 255, the maximum reflection value for the grayscale expression:(1)$$x_{normalized}=\frac{x-x_{min}}{x_{max}-x_{min}}$$

The normalized values were converted into a grayscale image. A value closer to zero indicates a closer distance between the plant and camera, and larger values indicate a farther distance between the plant and camera.

We also examined the characteristics of the dataset. Figure 3 depicts the normalized growth index values during the cultivation period of the entire dataset. As the crop grew, the height increased with a steeper slope than the diameter. As a result, the change rates of the fresh weight, dry weight, and leaf area increased in a quadratic equation, and the rates became higher toward the later growth stages. These growth patterns of the lettuce were consistent with previous studies [8,39].

### 2.2. Image Preprocessing

A total of 388 image pairs were used to build the model: 338 pairs for learning and 50 pairs for estimating unknown samples. A parallel input comprising RGB and depth images was generated for model training. First, the RGB images were cropped to 512 × 512 pixels from the center points of the plants so that the area information could be used for learning while maintaining the resolution of the input image to preserve the input pixel information. The center position of the plants was manually specified. For the depth images, the crop area was decreased to 256 × 256 pixels to minimize the effect of the background. The k-nearest neighbor method was applied to fill the pixels for which the depth image data were not acquired, and the average value of nine pixels was used [27,40].

All data were normalized to a value between zero and one through Equation (1). The RGB images were normalized using a maximum value of 255 and a minimum value of zero, which are the reflection values in the visible band. The depth images were normalized using a maximum value of 1306 and a minimum value of 681 from the measurements. The actual aboveground data used as labels were normalized to values between zero and one through Equation (1).

If data are insufficient for deep learning model training, errors increase greatly, leading to underfitting and poor model performance on the training and unknown datasets. In addition, overfitting may occur, deteriorating model performance on unknown data due to the large variance in the predicted values [41]. In this context, data augmentation was applied to sufficiently supplement the data. Data augmentation during learning involved rotating the original images at an arbitrary angle within 180°, horizontal inversion, and arbitrary image translation. The original train dataset of 338 pairs was extended to 119,868 pairs through augmentation.

### 2.3. Network Architecture and Training

A two-stage CNN model was proposed to obtain fresh weight, dry weight, height, diameter, and leaf area values for the input RGB and depth images (Figure 4). The model architecture was developed using the TensorFlow 2.5.1 framework and Python 3.8. The model structure consists of a model containing two individual models, unlike the MISO and MIMO models in the previous studies [27,28]. Two-stage architecture might be effective for data with time-series characteristics by connecting the two models in series [35,36,37,38]. In the first stage, the model performed learning and inference on fresh weight, dry weight, diameter, and height using RGB and depth images. The estimated fresh weight, diameter, and height were used as final model outputs. In the second stage, the results estimated in the previous model are used as input for a model to estimate the leaf area and correct the dry weight values.

In the first stage, transfer learning was applied to the CNN structure of the RGB image input model, and a pretrained model with the ImageNet dataset [42] was used. ResNet50V2 [43] (i.e., the improved structure of ResNet50 [44]) was used as the pretrained model architecture. When a CNN is trained through backpropagation, as the number of layers becomes deeper, the differential values for the loss between the target and predicted data decrease, and the weights of each layer filter that affect the model output are learned with small values. This phenomenon, called gradient vanishing, made the input data untrainable, consequently degrading model accuracy. However, ResNet solved the gradient vanishing problem by dividing each neural network layer into residual blocks and omitting each block through a skip connection. Moreover, ResNetV2 improved ResNet using the activation function in front of the convolutional layer in the residual block (Figure 5).

Two separate convolutional layers (3 × 3 and 1 × 1) were added before the pretrained model because the image size was larger than the original input size of the pretraining model. After the pretrained model was applied, an additional fully connected layer was placed to perform training suitable for the input dataset. We also tried to prevent overfitting by setting the dropout layers.

The architecture for the depth input model was designed similarly to the RGB model. However, without pretraining for ResNet50V2, the depth images were newly learned. The results of the RGB and depth input models were concatenated and passed through two fully connected layers with 2048 nodes and dropout layers.

In the second stage, the ANN model with two fully connected layers with 2048 nodes and dropout layers was deployed identically to the last part of the first stage to use the values inferred in the first stage for final growth prediction. Losses in all layers were calculated using the mean squared error measurements between the predicted value and actual label. Adam was used as an optimization function to minimize the result value of the loss function [45], and the rectified linear unit was applied as an activation function [46].

The model was trained using a desktop with Windows (Windows 10, Microsoft, Redmond, WA, USA), an Intel i7 central processing unit (CPU; i7-11700, Intel, Santa Clara, CA, USA), and a GeForce RTX GPU (RTX 3090, NVIDIA, Santa Clara, CA, USA) with 64 GB of RAM. The hyperparameter settings were as follows: The initial learning rate was set to 0.001, and 200-epoch learning was performed. If the validation loss did not decrease for eight iterations, the learning rate was halved by setting the learning rate scheduler. If the validation loss did not change for 30 iterations, early stopping was applied to reduce unnecessary learning time. The input batch size was set to 32. When the loss on the validation dataset became the smallest value, the model weights were preserved. In addition, to avoid overfitting from fixing a validation dataset, *k*-fold cross-validation was applied [47]. In *k*-fold cross-validation, a training dataset is divided into *k* pieces, and training is performed *k* − 1 times. For each training iteration, training is performed with *k* − 1 datasets, and one dataset is used for validation. This study applied the commonly used five-fold cross-validation [48,49].

### 2.4. Model Evaluation

The growth indices were estimated using a testing dataset not used for training, and the R^2^, RMSE, and NRMSE between the estimated and actual values were analyzed to evaluate the estimation accuracy of the developed model during the entire cultivation period. The NRMSE was calculated using Equation (2) with the maximum and minimum values for the entire dataset.

Demonstrating the effectiveness of the developed model in the late growth period can improve model applicability. The training results using only RGB images were compared with those using RGB and depth images in 20 samples of late growth. We analyzed an overall change of five growth indices using Equations (1) and (3), and the change increased rapidly in the later 20 images:(2)$$\mathrm{NRMSE}=\frac{\mathrm{RMSE}}{x_{max}-x_{min}}$$
(3)$$\begin{array}{c} Normalized\;growth\;index\;effect=\prod Growth\;index_{normalized}\\ Growth\;index=\left\{Fresh\;weight,\;Dry\;weight,\;Height,Diameter,Leaf\;area\right\} \end{array}$$

In addition, to numerically compare the performance with a previous study [28], the NMSE value was calculated using Equation (4):(4)$$\mathrm{NMSE}=\overset{m}{\underset{j=1}{\sum }}\frac{\overset{n}{\underset{i=1}{\sum }}{\left(gt_{ij}-p_{ij}\right)}^{2}}{\overset{n}{\underset{i=1}{\sum }}{\left(gt_{ij}\right)}^{2}}$$
where *gt* is the actual growth indices, *p* is the estimated value, *n* is the number of images, and *m* is the number of growth indices.

The loading and inference times per image were measured on the desktop for training, and an embedded board (Jetson SUB mini-PC, Nanshan, Shenzhen, China) based on a Jetson Xavier (Jetson Xavier NX, NVIDIA, Santa Clara, CA, USA) module was used to evaluate the real-time image processing performance of the developed model. The Jetson SUB mini-PC was operated on an Ubuntu 18.04 operating system equipped with a 64-bit Carmel ARM (Carmel v8.2, NVIDIA, Santa Clara, CA, USA) CPU, and a Volta GPU (Volta, NVIDIA, Santa Clara, CA, USA) with 8 GB of RAM.

## 3. Results and Discussion

Figure 6 illustrates the changes in training loss and validation loss while training the first and second model stages. In the first and second stages, the validation loss tends to decrease as the training loss decreases. This result revealed that the gap between the training loss and validation loss narrowed. The model weight was not saved or passed on to the next training with another validation fold using the best checkpoint when the validation loss did not converge in every cross-validation cycle to prevent overfitting, where the validation loss diverges and increases.

In deep learning applications, checkpointing operates similarly to early stopping [50], and early stopping is one of the representative strategies to prevent overfitting, although checkpointing does not necessarily eliminate overfitting [51].

Figure 7 and Table 3 compare the lettuce growth indicators estimated by the developed model using RGB-D and the actual measured values. The results reveal an R^2^ greater than 0.88 and a slope greater than 0.90 for all growth indices, implying that the developed CNN model built using RGB-D images could estimate with considerable accuracy.

Similar to previous studies, the image interpretation capability of the CNN made it possible to implement the model with a good explanatory power [22,28]. The R^2^ and NRMSE values for the leaf area were better than those of the other indices. Depending on the plant growth, the point at which each leaf is attached to the stem and the angle of each leaf may vary. The leaf positions and angles contain information in the vertical direction and can affect the nonprojected leaf area. Fresh and dry weights are also affected when plants grow vertically. The R^2^ and NRMSE values of the fresh and dry weights support this. Therefore, an RGB-D image model may be effective for estimating growth indices.

In a previous study [28], Raja et al. proposed a CNN-based estimation model using the same dataset as that in this study [30]. The NMSE value of this study was 0.87, larger than the NMSE values of 0.068 and 0.069 of the MIMO and MISO models to which the DCNN was applied in a previous study. However, the NMSE for this model was better than the values of 0.092 and 0.088 for the MIMO and MISO models using the original CNN. Compared to the RMSE values of the MIMO model using the CNN by Raja et al. (fresh weight: 29.95, dry weight: 1.1, height: 1.75, diameter: 2.3, and leaf area: 344.96), the results (Table 3) were slightly improved except for dry weight. In contrast, the proposed model presented lower accuracy for the fresh weight, dry weight, and leaf area than the results of applying the DCNN to MIMO (fresh weight: 20.54, dry weight: 0.98, height: 1.75, diameter: 2.57, and leaf area: 274.23). The MISO model using the DCNN exhibited the best RMSEs for the growth indices except for fresh weight (fresh weight: 25.53, dry weight: 0.88, height: 1.64, diameter: 2.39, and leaf area: 272.58). The overall results of SIMO and SISO models using CNN and DCNN were similar to the proposed model results, with high and low indices.

However, our study focused on investigating the effectiveness of using RGB-D images and the CNN in the late growth stage rather than improving the accuracy. Securing the real-time applicability of the developed model was also a major concern. Therefore, given this real-time applicability, the multi-output model was developed and investigated, and the single-output model was not considered.

The developed RGB-D model also reduced errors in the late growth stage of lettuce, as displayed in Figure 8, which presents normalized deviations between the actual values and indices estimated from the newly developed model using RGB-D images and only RGB images in 20 samples of the late growth. Depth processing layers were skipped in the two-stage model to obtain the results using only RGB images. There was no significant difference between the performance of the developed model without the depth processing layers and the original ResNetV50 model using RGB images.

In Figure 8, the deviations were improved compared to the RGB-only model in the late growth, except for the diameter. These results indicated that using depth images in the CNN could advance the estimation performance for fresh weight, dry weight, height, diameter, and leaf area in the late growth stage. The increased accuracy for the late growth stage may lead to model applicability over the entire growth cycle. Figure 3 presents the quadratic increase in fresh weight and leaf area by vertical growth of crops. Due to this growth pattern, 3D information, such as depth images, may improve the growth estimation model.

In contrast, model accuracy using only RGB images was relatively higher in the case of diameter. The result may be because the diameter is not significantly affected by the vertical information of the crops, as the diameter was measured in the projected plant area.

In addition to the growth cycle, for the growth indices for each lettuce cultivar, the R^2^ values are similar to the estimated results for all cultivars (Table 4). Therefore, it is expected to be widely applicable to various varieties, such as *Lactuca sativa* L., domestic lettuce grown in South Korea.

Table 5 lists the average time and standard deviations required to load one image and infer growth indicators on a desktop and embedded board. The inference speed was still reasonable for the RGB-D images and the CNN model. The average loading and inference time per image was 0.83 s on the embedded board, which is a slightly longer time than that on the desktop. An on-the-go image acquisition device presented in a previous study [9] required approximately 2.83 s to capture one image per plant and was driven by a motor to move to the next plant. In an on-the-go image acquisition system, the time for image cropping may not be considered because the camera is located at the center of the plant. The processing time presented in this study corresponds to approximately 31% of the device running time; therefore, it is likely to be sufficiently applicable to real-time image acquisition and estimation via an on-the-go device through multiprocessing.

## 4. Conclusions

In this study, we developed and evaluated a two-stage CNN model using RGB and depth images to estimate the fresh weight, dry weight, height, diameter, and leaf area of lettuce grown in a greenhouse hydroponics system. The RGB-D images and CNN-based architecture effectively estimate the growth indices of greenhouse lettuces, demonstrating that the CNN-based model using RGB-D images may provide nondestructive, fast, and accurate measurements of growth indices.

The proposed RGB-D model can reduce estimation errors in the late growth phase when the growth indices are greatly increased. This advantage can improve model applicability for estimating growth indices. The model can potentially effectively model the interaction between crop physiology and the environment.

Since the late stage of crop growth is closely related to the optimal harvest time, the proposed model may also help predict the proper harvest time of crops and develop decision support systems or harvesting robots that operate automatically. The rapid inference speed of the CNN on individual images also enables sensing and harvesting tasks requiring real-time motion and image processing.

The lettuce images used in this study include a limited number of varieties and a small volume of data. In addition, data were collected manually. In the future, we will apply the developed model to an on-the-go embedded system in a greenhouse and collect and learn from a large volume of data from lettuce varieties grown in South Korea to perform real-time growth estimation and improve the accuracy and measurement convenience. Another goal for improvement concerns the automatic measurement of plant center points.

Although this study was conducted only on lettuce, the CNN model is more versatile than traditional image processing methods and can be applied to various environments and crops after slight tuning, such as through transfer learning. The proposed estimation model using RGB-D data can be applied to fruit and vegetable (in addition to leafy vegetable) crops.

The rapid and precise measurement of growth indicators using RGB-D images and CNN models has great potential for agricultural applications. Developing such estimation models will advance precision agriculture and ultimately improve the productivity of horticultural crops.

## Figures and Tables

**Figure 1 sensors-22-05499-f001:**
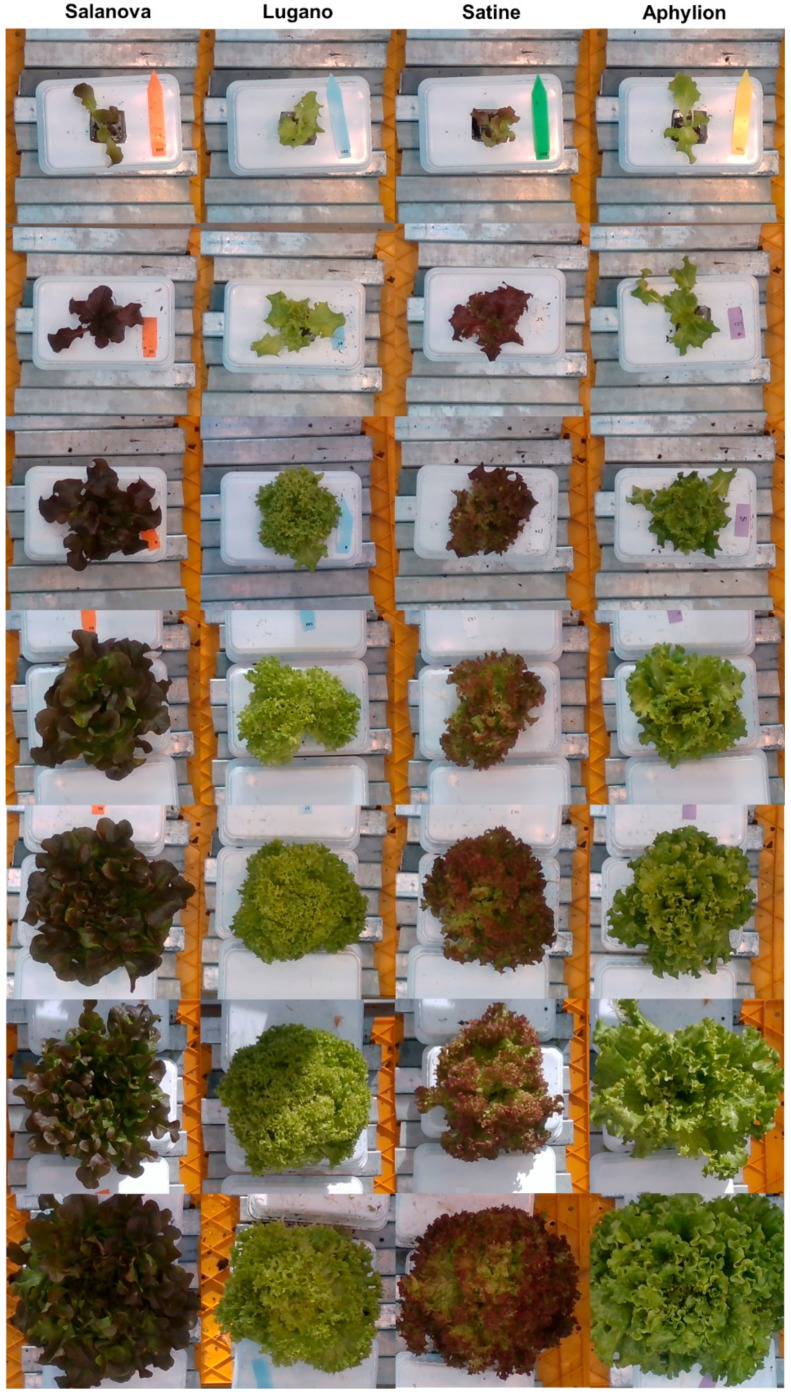
RGB images of sampled Salanova, Lugano, Satine, and Aphylion cultivar lettuce grown for seven weeks (growing stages at one-week intervals after transplanting). The images are cropped to 512 × 512 pixels from the center. The dataset is available in online: https://doi.org/10.4121/15023088.v1 [30].

**Figure 2 sensors-22-05499-f002:**
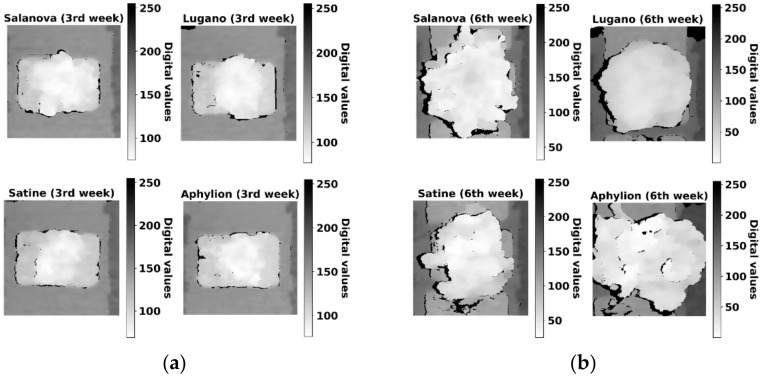
Normalized depth images with grayscale to visualize digital values from 681 to 1306 (high: close to the camera, low: far from the camera): (**a**) depth images of the four lettuce varieties 3 weeks after transplanting; and (**b**) depth images of the four lettuce varieties 6 weeks after transplanting.

**Figure 3 sensors-22-05499-f003:**
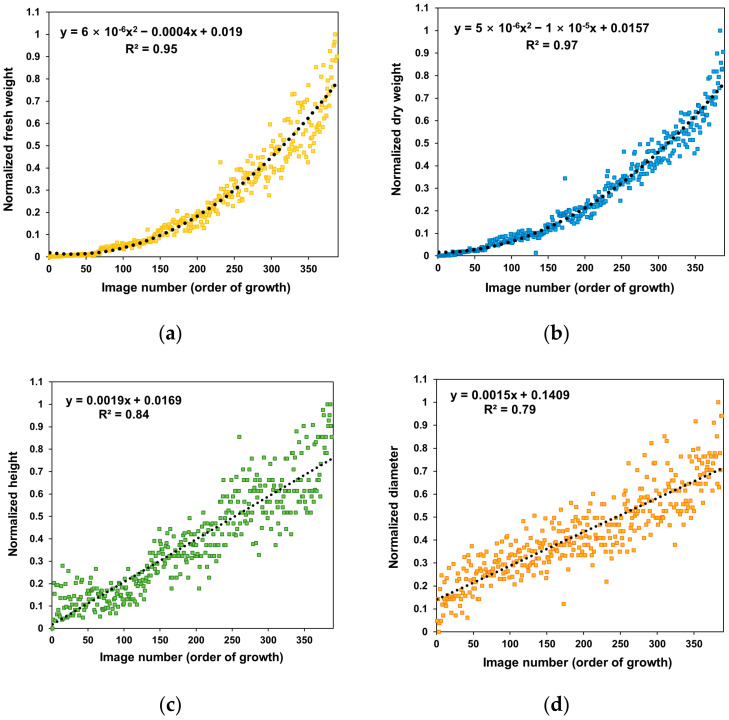
Change rates of growth indices in the dataset according to the crop growth ordered as the largest normalized growth index effect obtained from the product of each index: (**a**) fresh weight, (**b**) dry weight, (**c**) height, (**d**) diameter, and (**e**) leaf area.

**Figure 4 sensors-22-05499-f004:**
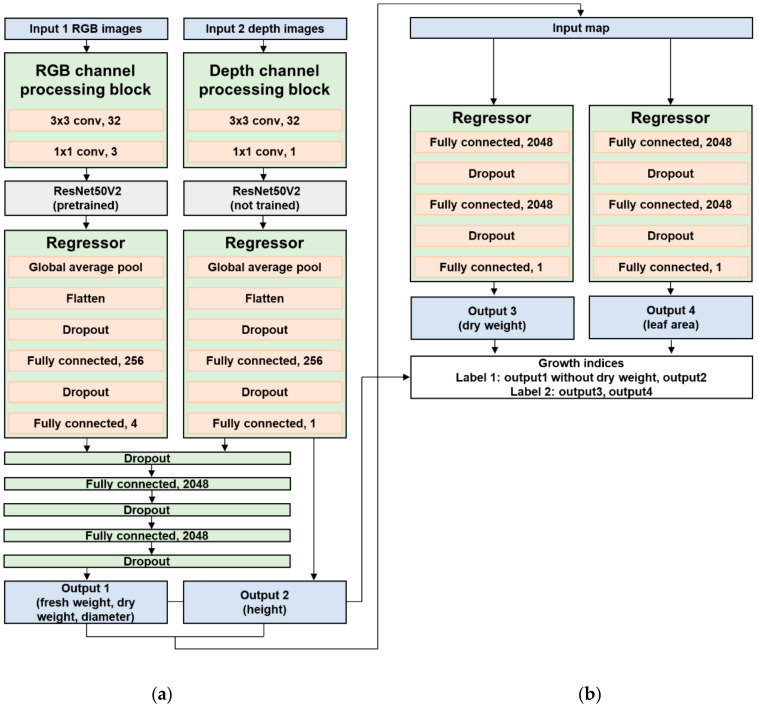
Two-stage architecture. (**a**) First stage: parallel RGB and depth input models based on untrained ResNet50V2 to estimate fresh weight, dry weight, diameter, and height. (**b**) Second stage: parallel artificial neural network regression model to estimate the dry weight and leaf area.

**Figure 5 sensors-22-05499-f005:**
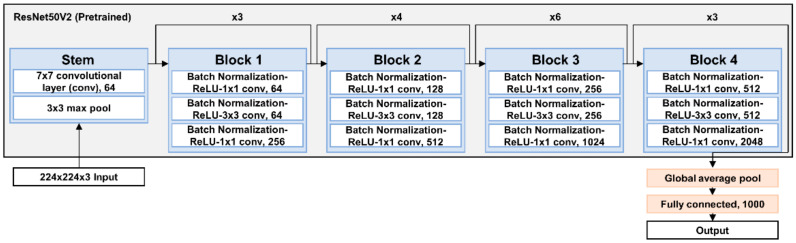
CNN architecture of ResNet50V2, which was used as backbone model of presented model.

**Figure 6 sensors-22-05499-f006:**
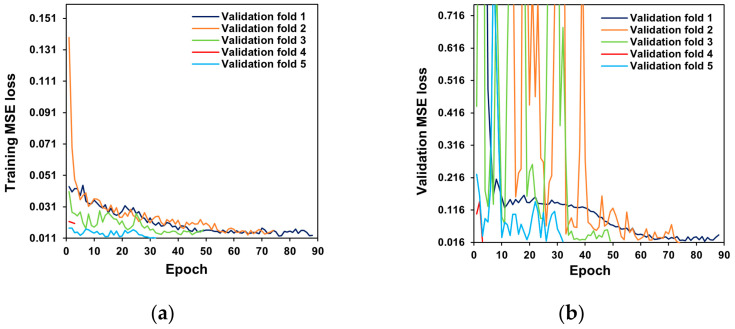
Change in training loss and validation loss during training of the developed model: (**a**) training loss in the first stage, (**b**) validation loss in the first stage, (**c**) training loss in the second stage, and (**d**) validation loss in the second stage. Repetition of loss reduction depicts five-fold cross-validation. The model weight was not kept when the validation loss did not converge in every fold to avoid overfitting.

**Figure 7 sensors-22-05499-f007:**
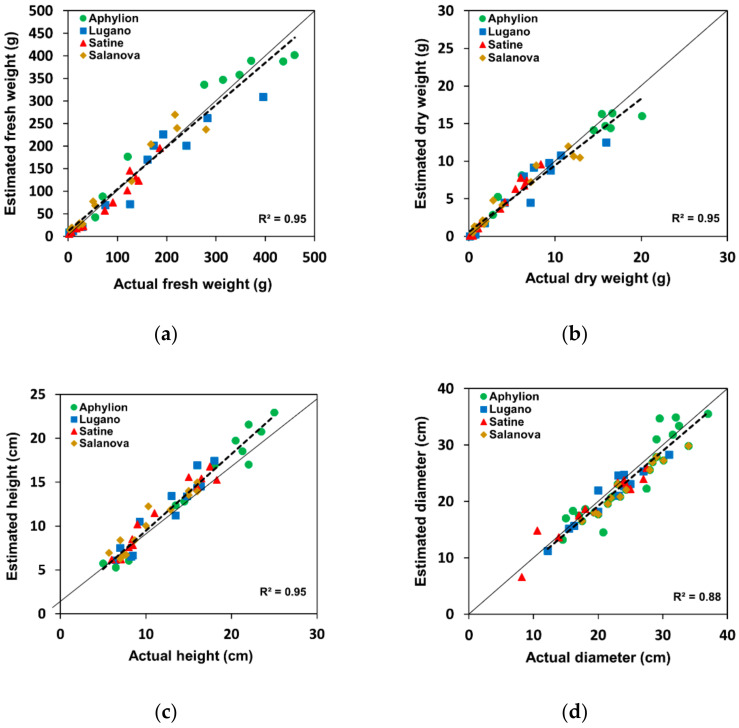
Comparison of growth indicators estimated from the developed model using RGB-D and those measured using the destructive method. The color indices present the values for each lettuce variety: (**a**) fresh weight, (**b**) dry weight, (**c**) height, (**d**) diameter, and (**e**) leaf area.

**Figure 8 sensors-22-05499-f008:**
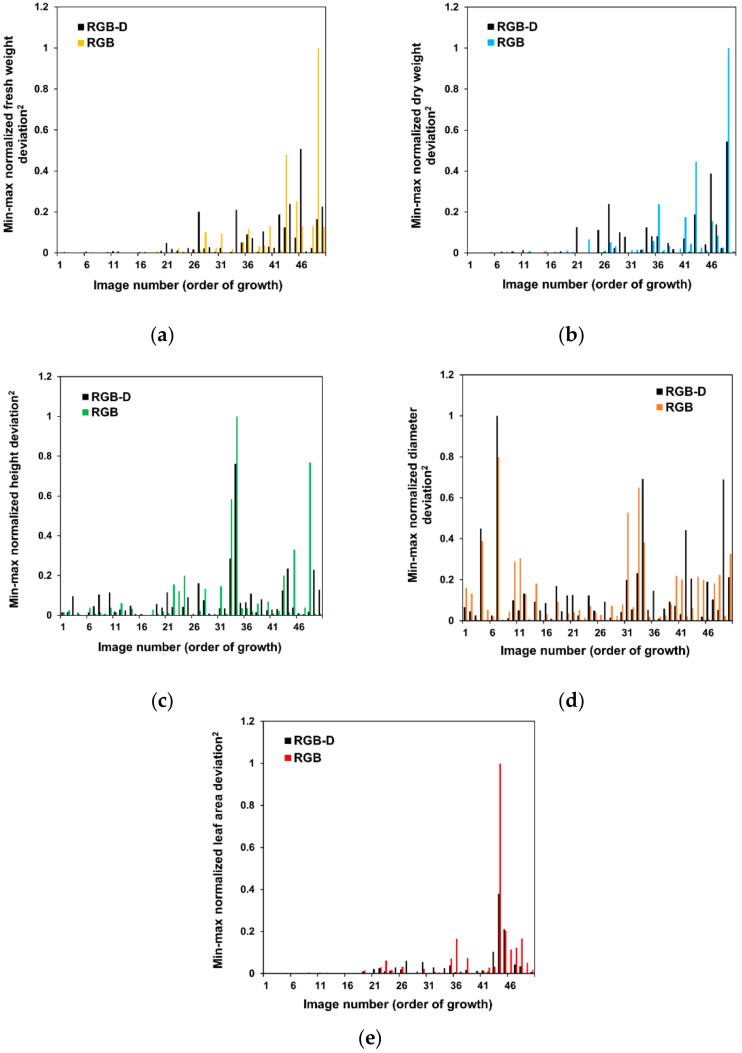
Comparison of growth index estimation of the developed two-stage CNN model using RGB-D images and RGB images in 20 samples in the late growth stage: (**a**) fresh weight, (**b**) dry weight, (**c**) height, (**d**) diameter, and (**e**) leaf area.

**Table 1 sensors-22-05499-t001:** Growth index data format for each RGB image of the lettuce measured using destructive methods.

Week	RGB-Image(Salanova)	Fresh Weight (g)	Dry Weight (g)	Height(cm)	Diameter(cm)	Leaf Area(cm^2^)
1	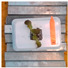	5.2	0.58	9.8	17.2	202.7
2	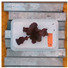	16.4	1.22	6.8	18.5	520.1
3	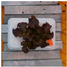	69.8	4.16	9.0	25.1	1694.3
4	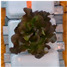	85.0	5.02	8.0	25.0	2008.8
5	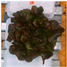	110.0	6.04	12.0	28.0	2414.7
6	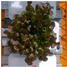	133.2	6.84	15.8	32.0	3089.2
7	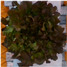	236.5	11.04	17.0	33.5	5348.1

**Table 2 sensors-22-05499-t002:** Maximum and minimum values for each growth index within the dataset measured using destructive methods.

	Fresh Weight (g)	Dry Weight (g)	Height (cm)	Diameter (cm)	Leaf Area (cm^2^)
Minimum	1.4	0.09	4.3	8.2	57.6
Maximum	459.7	20.1	25	37	5868

**Table 3 sensors-22-05499-t003:** Growth index estimation of the developed two-stage CNN model using RGB-D images.

	Fresh Weight (g)	Dry Weight (g)	Height (cm)	Diameter (cm)	Leaf Area (cm^2^)
R^2^	0.95	0.95	0.95	0.89	0.96
RMSE ^1^	27.85	1.26	1.53	2.28	326.04
NRMSE ^2^ (%)	6.09	6.30	7.65	7.92	5.62

^1^ Root mean square error. ^2^ Normalized root mean square error. Ratio of RMSE to the difference between the maximum and minimum value for the entire dataset.

**Table 4 sensors-22-05499-t004:** Comparison of R^2^ values of four lettuce varieties.

	Fresh Weight (R^2^)	Dry Weight (R^2^)	Height (R^2^)	Diameter (R^2^)	Leaf Area (R^2^)
Aphylion	0.96	0.96	0.96	0.92	0.96
Lugano	0.93	0.91	0.91	0.86	0.96
Satine	0.96	0.99	0.95	0.92	0.98
Salanova	0.95	0.95	0.91	0.96	0.97
Total	0.95	0.95	0.95	0.89	0.96

**Table 5 sensors-22-05499-t005:** Processing time required to load and infer lettuce RGB-D images using an i7-11700 CPU and GeForce RTX-3090 GPU versus a Jetson SUB mini-PC based on Jetson Xavier NX.

	Loading Time per Image (s)	Loading Time per 50 Images (s)	Inference Time per Image (s)	Inference Time per 50 Images (s)
i7-11700 and RTX-3090 PC	0.03 ± 0.001	0.52 ± 0.015	0.13 ± 0.044	0.55 ± 0.079
Jetson SUB mini-PC	0.034 ± 0.004	2.56 ± 0.990	0.49 ± 0.190	5.59 ± 0.108

## Data Availability

This study used the Third Autonomous Greenhouse Challenge: Online Challenge Lettuce Images dataset publicly available at 4TU.ResearchData [30]. The ImageNet dataset was also used, which is available on the ImageNet website [42]. More details about the data are available in Section 2.1, Section 2.2 and Section 2.3.

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
