# Peer review of "Estimation of Greenhouse Lettuce Growth Indices Based on a Two-Stage CNN Using RGB-D Images"

_sensors, 2022, doi:10.3390/s22155499_

Round 1

Reviewer 1 Report

In this paper, a two-stage CNN architecture is designed for estimating the growth indices of greenhouse lettuce using RGB-D images. The authors spent paragraphs for introducing the existing dataset [30] (p.2 – p.6), and a general image normalization skill in Eq. (1) (for visualization) and Sec. 2.2. In Sec. 2.3, the authors adopted the famous CNN-based architecture, ResNet50V2 [36] (Fig. 6), to the proposed two-stage architecture (Fig. 5). In Sec. 2.3, the classical NRMSE Eq. (2) is used for model evaluation, and the experimental results are shown in Table 2 and Table 3. Moreover, the execution time for a RTX-3090 pc and a Jetson SUB Mini PC are shown in Table 4. The authors should read and have a comparison to this paper:

[1]P. Raja, A. Olenskyj, H. Kamangir, M. Earles, “Simultaneously Predicting Multiple Plant Traits from Multiple Sensors via Deformable CNN Regression”, arXiv:2112.03205v1 [cs.CV] 6 Dec 2021

[2] Jinsong Li, Xiyue Guo, Ying Wang, Maowei Li, Lihua Zheng, Minjuan Wang, “An End-to-End Deep RNN based Network Structure to Precisely Regress the Height of Lettuce by Single Perspective Sparse Point Cloud”, https://www.cs.usask.ca/faculty/stavness/cvppa2021/abstracts/Li_52.pdf

Especially the Table 1 of [1].

In addition, the following points should be addressed:

1.     Since [27] proposed a two-stream CNN model, what is the difference from that paper to the proposed method (two-stage)? A proper property comparison and performance comparison are necessary.

2.     Should the authors have proper comparison to [8] [28][29]?

3.     In line 175-Line 178 of p.7, how many images do you have after data augmentation?

4.     In Line 226- Line 230 of p. 9, the authors should show the results of epoch vs. loss and epoch vs. accuracy.

5.     Why did the authors use five-fold cross-validation? Is there any reason?

Author Response

Dear Editor and Reviewers,

We would like to thank you and the reviewers for the careful comments and suggestions to improve the quality of our manuscript. We have tried our best in revising the manuscript to address all of the issues raised by the reviewers. The manuscript revised according to the comments and edits has been uploaded. Here is a listing of the reviewer comments along with our responses to the comments. Line notation in this letter was made based on the screen showing all changes in the tracking function of MS WORD.

Reviewer #1

Comments and Suggestions for Authors

In this paper, a two-stage CNN architecture is designed for estimating the growth indices of greenhouse lettuce using RGB-D images. The authors spent paragraphs for introducing the existing dataset [30] (p.2 – p.6), and a general image normalization skill in Eq. (1) (for visualization) and Sec. 2.2. In Sec. 2.3, the authors adopted the famous CNN-based architecture, ResNet50V2 [36] (Fig. 6), to the proposed two-stage architecture (Fig. 5). In Sec. 2.3, the classical NRMSE Eq. (2) is used for model evaluation, and the experimental results are shown in Table 2 and Table 3. Moreover, the execution time for a RTX-3090 pc and a Jetson SUB Mini PC are shown in Table 4. The authors should read and have a comparison to this paper:

[1]P. Raja, A. Olenskyj, H. Kamangir, M. Earles, “Simultaneously Predicting Multiple Plant Traits from Multiple Sensors via Deformable CNN Regression”, arXiv:2112.03205v1 [cs.CV] 6 Dec 2021

[2] Jinsong Li, Xiyue Guo, Ying Wang, Maowei Li, Lihua Zheng, Minjuan Wang, “An End-to-End Deep RNN based Network Structure to Precisely Regress the Height of Lettuce by Single Perspective Sparse Point Cloud”, https://www.cs.usask.ca/faculty/stavness/cvppa2021/abstracts/Li_52.pdf

Especially the Table 1 of [1].

  • Thanks for letting us know about the two papers. We did not know the papers were previously reported based on the use of the same dataset as ours. In the revisions, we have included the citations of the two papers in the introductory section by describing the main results (lines 83-89). In addition, a comparison has been made by mentioning what were different from our research (lines 90-102) that includes the evaluation of real-time applicability and the use of both RGB and depth data for the entire growth cycle of lettuce.

In addition, the following points should be addressed:

  1. Since [27] proposed a two-stream CNN model, what is the difference from that paper to the proposed method (two-stage)? A proper property comparison and performance comparison are necessary.

  • We have added sentences to more clearly describe the difference between the two-stream and two-stage models (lines 78-79, 99-102, and 211-214). For example, as reported in a previous study (reference 27), the two-stream model was developed based on a multi-input single-output (MISO) structure. On the other hand, the two-stage model used in the research was built based on the generation of multiple outputs by connecting two separate models in series.

  1. Should the authors have proper comparison to [8][28][29]?

  • We have deleted the reference [8] because it turned out that the ref. 8 was not so closely relevant to references 28 and 29
  1. In line 175-Line 178 of p.7, how many images do you have after data augmentation?

  • We have added information about the image number after data augmentation to our manuscript (lines 199-200). Thank you.

  1. In Line 226- Line 230 of p. 9, the authors should show the results of epoch vs. loss and epoch vs. accuracy.

  • Following the suggestion, we have added a figure (Fig. 8) and a related paragraph (lines 276-286) to report the results of epoch vs. loss and epoch vs. accuracy. Thank you.

  1. Why did the authors use five-fold cross-validation? Is there any reason?

  • Since the reason of using the five-fold cross-validation was because it has been proven to be commonly effective from previous studies in the agricultural area, we have mentioned the point in the manuscript (line 260).

Reviewer 2 Report

(1) Using camera to determine weight of vegetables is not new, the problem was how to determine vegetable density and the distance between camera and object.  

(2)3D is better than 2D. Two camera is better than one camera for determine distance and TRUE volume.  

Author Response

Dear Editor and Reviewers,

We would like to thank you and the reviewers for the careful comments and suggestions to improve the quality of our manuscript. We have tried our best in revising the manuscript to address all of the issues raised by the reviewers. The manuscript revised according to the comments and edits has been uploaded. Here is a listing of the reviewer comments along with our responses to the comments. Line notation in this letter was made based on the screen showing all changes in the tracking function of MS WORD.

Reviewer #2

Comments and Suggestions for Authors

(1) Using camera to determine weight of vegetables is not new, the problem was how to determine vegetable density and the distance between camera and object.

  • We agree with the opinion that previous studies used a camera to determine the vegetable weights without the consideration of vegetable density and distance information. On the other hand, since our research focuses on the use of depth information obtained with the RealSense camera to obtain 3D information, we would think that it can be related to the novelty of our research. To more clearly emphasize the point in the manuscript, we have revised the introductory section (lines 45-47). In addition, as part of what is different from previous studies, a sentence has been revised that describes the evaluation of real-time applicability for the developed algorithm (lines 90-98) so that readers can be convinced that our research has two main points, i.e., use of depth information and evaluation of real-time applicability in terms of novelty.

(2)3D is better than 2D. Two camera is better than one camera for determine distance and TRUE volume.

  • We agree with the reviewer’s opinion. That’s why we used depth images to improve the estimation performance of fresh weights for lettuces based on the use of camera images obtained during the entire growth period, because a change in lettuce volume over time is not linear, thereby requiring the use of 3D data. To emphasize the point, we have revised a sentence (lines 90–98) to justify the necessity for the collection of depth information .

Reviewer 3 Report

No comment

Author Response

Dear Editor and Reviewers,

We would like to thank you and the reviewers for the careful comments and suggestions to improve the quality of our manuscript. We have tried our best in revising the manuscript to address all of the issues raised by the reviewers. The manuscript revised according to the comments and edits has been uploaded. Here is a listing of the reviewer comments along with our responses to the comments. 

Reviewer #3

No comments.

We have received additional English proofreading from a native speaker.

Thank you for giving us the opportunity to improve the quality of our paper.

Sincerely yours.

Round 2

Reviewer 1 Report

In the revision of this paper, what I mentioned in the previous review,

1.     [1]P. Raja, A. Olenskyj, H. Kamangir, M. Earles, “Simultaneously Predicting Multiple Plant Traits from Multiple Sensors via Deformable CNN Regression”, arXiv:2112.03205v1 [cs.CV] 6 Dec 2021

Especially the Table 1 of [1].

No experimental comparison to that Table is provided. Moreover, no proper citation is properly shown in the revision of the article. Is the reference number  [30] or [28]? In line 90-102, no experimental results are provided.

2.     Should the authors have proper comparison to [8][28][29]?

The authors just remove the comparison [8], but no proper comparison (experimental results) to [28] and [29] are not provided.

3.     In line 175-Line 178 of p.7, how many images do you have after data augmentation?

The authors replies: “We have added information about the image number after data augmentation to our manuscript (lines 199-200). 

However, there are no image number information provided in Line 199-200.

4.     In Line 226- Line 230 of p. 9, the authors should show the results of epoch vs. loss and epoch vs. accuracy.

 The authors replies:”Following the suggestion, we have added a figure (Fig. 8) and a related paragraph (lines 276-286) to report the results of epoch vs. loss and epoch vs. accuracy. Thank you.”

However, there is no epoch vs. loss and epoch vs. accuracy information provided in Fig. 8. Do you mean Fig. 7? In Fig. 7 the overfitting issue happened. The authors should deal with the overfitting issue in the proposed method.

The authors should carefully proofread the entire article, and the current method (machine learning model) is not stable (overfitting issue not dealt with). 

Author Response

Dear Editor and Reviewers,

We would like to thank you for the careful comments and suggestions to improve the quality of our manuscript. We have tried to revise the manuscript to address all of the issues raised by the reviewers. In addition, we have received English proofreading from an official English editing service one more time as attached. Here is a uploading of our responses and supplementary materials to the comments. We have written line notation based on the screen showing all changes in the tracking function of MS WORD (all markup option).

Sincerely yours.

Hak-jin Kim, Ph.D.

Seoul National University

Round 3

Reviewer 1 Report

Most of my raised points are dealt with in the revised version of the paper.